# Robust upward dispersion of the neutron spin resonance in the heavy fermion superconductor $Ce_{1-x}Yb_xCoIn_5$

Yu Song[1], John Van Dyke[2], I.K. Lum[3,4,5], B.D. White[4,5], Sooyoung Jang[3,4,5], Duygu Yazici[3,4,5], L. Shu[6,7], A. Schneidewind[8], Petr Čermák[8], Y. Qiu[9], M.B. Maple[3,4,5], Dirk K. Morr[2] & Pengcheng Dai[1]

The neutron spin resonance is a collective magnetic excitation that appears in the unconventional copper oxide, iron pnictide and heavy fermion superconductors. Although the resonance is commonly associated with a spin-exciton due to the $d(s^{\pm})$-wave symmetry of the superconducting order parameter, it has also been proposed to be a magnon-like excitation appearing in the superconducting state. Here we use inelastic neutron scattering to demonstrate that the resonance in the heavy fermion superconductor $Ce_{1-x}Yb_xCoIn_5$ with $x = 0$, 0.05 and 0.3 has a ring-like upward dispersion that is robust against Yb-doping. By comparing our experimental data with a random phase approximation calculation using the electronic structure and the momentum dependence of the $d_{x^2-y^2}$-wave superconducting gap determined from scanning tunnelling microscopy (STM) for $CeCoIn_5$, we conclude that the robust upward-dispersing resonance mode in $Ce_{1-x}Yb_xCoIn_5$ is inconsistent with the downward dispersion predicted within the spin-exciton scenario.

[1] Department of Physics and Astronomy, Rice University, Houston, Texas 77005, USA. [2] Department of Physics, University of Illinois at Chicago, Chicago, Illinois 60607, USA. [3] Materials Science and Engineering Program, University of California, San Diego, La Jolla, California 92093, USA. [4] Department of Physics, University of California, San Diego, La Jolla, California 92093, USA. [5] Center for Advanced Nanoscience, University of California, San Diego, La Jolla, California 92093, USA. [6] State Key Laboratory of Surface Physics, Department of Physics, Fudan University, Shanghai 200433, China. [7] Collaborative Innovation Center of Advanced Microstructures, Nanjing 210093, China. [8] Jülich Center for Neutron Science JCNS, Forschungszentrum Jülich GmbH, Outstation at MLZ, D-85747 Garching, Germany. [9] NIST Center for Neutron Research, National Institute of Standard and Technology, Gaithersburg, Maryland 20899, USA. Correspondence and requests for materials should be addressed to D.K.M. (email: dkmorr@uic.edu) or to P.D. (email: pdai@rice.edu).

Understanding the origin of unconventional super-conductivity in strongly correlated electron materials continues to be at the forefront of modern condensed matter physics[1–5]. In copper oxide[6–8], iron pnictide[9,10] and heavy fermion[11,12] superconductors, the appearance of a neutron spin resonance below the superconducting transition temperature $T_c$ suggests that spin-fluctuation-mediated pairing is a common thread for different families of unconventional superconductors[2].

The neutron spin resonance is a collective magnetic excitation coupled to superconductivity with a temperature dependence similar to the superconducting order parameter[6,7]. It is located near the antiferromagnetic (AF) ordering wave vector $\mathbf{Q}_{AF}$ of the undoped parent compound and its energy $E_r$ at $\mathbf{Q}_{AF}$ is related to either $T_c$ (ref. 13) or the superconducting energy gap $\Delta$ (ref. 14). Although it is generally accepted that the resonance is a signature of unconventional superconductivity[2], there is no consensus on its microscopic origin. A common interpretation of the resonance is that it is a spin-exciton, arising from particle-hole excitations involving momentum states near the Fermi surfaces that possess opposite signs of the $d$ (or $s^{\pm}$)-wave superconducting order parameter[7,12,15]. Alternatively, it has also been proposed to be a magnon-like excitation[16,17]. At present, there is no consensus on its microscopic origin[2,7,8,10].

In hole-doped copper oxide superconductors, the magnetic excitations have an hourglass dispersion with a downward dispersion at energies below $E_r$ and an upward magnon-like dispersion at energies above $E_r$ (ref. 8). The resonance, on the other hand, obtained by subtracting the normal-state magnetic excitations from those in the superconducting state, displays predominantly a downward dispersion[18–21]. In the case of Ni-underdoped BaFe$_2$As$_2$ with coexisting AF order and superconductivity[22], the resonance only reveals an upward magnon-like dispersion[23]. In both cases, the resonance is well described by the spin-exciton scenario, the opposite dispersions being a result of $d_{x^2-y^2}$ or $s^{\pm}$ symmetry of the superconducting order parameter[23,24].

For the heavy fermion superconductor CeCoIn$_5$ ($T_c = 2.3$ K) (ref. 4), the resonance appears below $T_c$ at $E_r = 0.60 \pm 0.03$ meV and the commensurate AF wave vector $\mathbf{Q}_{AF} = (1/2, 1/2, 1/2)$ in reciprocal space[12]. Since CeCoIn$_5$ has a superconducting gap with $d_{x^2-y^2}$-wave symmetry as determined from scanning tunnelling microscopy (STM) experiments[25,26], the resonance is expected to show a downward dispersion similar to the cuprates within the spin-exciton picture[27,28]. Alternatively, the resonance, with its three-dimensional character[12], could be a magnon-like excitation of $f$ electrons that becomes visible due to its reduced decay rate in the superconducting state[16,17]. In this case, the resonance is not a signature of $d_{x^2-y^2}$-wave superconductivity, but a measure of the hybridization between $f$ electrons and conduction electrons and its associated pairing-sensitive Landau damping[17].

When La is substituted for Ce in Ce$_{1-x}$La$_x$CoIn$_5$ (refs 29,30), superconductivity and the energy of the resonance are both rapidly suppressed, while $E_r/k_B T_c$ remains approximately constant, where $k_B$ is the Boltzmann constant. At the same time, the energy width of the resonance broadens with increasing La-doping[31,32]. When Yb is doped into CeCoIn$_5$ to form Ce$_{1-x}$Yb$_x$CoIn$_5$, superconductivity is suppressed much slower[33]. With increasing Yb, de Haas-van Alphen and angle-resolved photo-emission spectroscopy studies find a change in the Fermi-surface topology between Yb nominal doping levels of $x = 0.1$ and $0.2$ (refs 34,35). In addition, London penetration depth measurements suggest that the superconducting gap changes from nodal to nodeless around a similar Yb-doping level[36], arising possibly from composite electron pairing in a fully gapped superconductor for $x > 0.2$ (ref. 37). If the resonance in CeCoIn$_5$ is a spin-exciton, it should be dramatically affected by the Yb-doping-induced changes in Fermi surface topology and superconducting gap. On the other hand, if the resonance is a magnon-like excitation, it should be much less sensitive to Yb-doping across $x = 0.2$ and display a upward dispersion similar to spin waves in antiferromagnetically ordered nonsuperconducting CeRhIn$_5$ characteristic of a robust effective nearest-neighbour exchange coupling, regardless of its itinerant electron or local moment origin[7,38,39].

Here we use inelastic neutron scattering to demonstrate that the resonance in the heavy fermion superconductor Ce$_{1-x}$Yb$_x$CoIn$_5$ with $x = 0$, 0.05 and 0.3, and $T_c \approx 2.3$, 2.25 and 1.5 K, respectively (Methods section and Supplementary Fig. 1)[4,12,33], has a dominant ring-like upward dispersion that is robust against Yb-doping and the concomitant changes in electronic structure, a feature not present in the spin-exciton scenario. Moreover, a downward dispersion expected in the spin-exciton scenario is not observed. The robust upward dispersion of the resonance suggests that it may have a magnon-like contribution[17]. Specifically, we find that the resonance in Ce$_{0.95}$Yb$_{0.05}$CoIn$_5$ displays an upward dispersion along [$H$, $H$, 0.5], [0.5, $K$, 0.5] and [0.5, 0.5, $L$] as shown in Fig. 1d–f, respectively. Upon increasing Yb-doping to $x = 0.3$, the energy of the resonance at $\mathbf{Q}_{AF}$ decreases corresponding to the reduction in $T_c$ (Supplementary Fig. 2), but the overall dispersion and location of the mode in reciprocal space remain unchanged. Upward dispersions similar to Ce$_{0.95}$Yb$_{0.05}$CoIn$_5$ are also found in undoped CeCoIn$_5$ and Ce$_{0.7}$Yb$_{0.3}$CoIn$_5$ (Supplementary Figs 3–5). Using the electronic structure and the momentum dependence of the $d_{x^2-y^2}$-wave superconducting gap determined from STM for CeCoIn$_5$ (Fig. 1g)[28], we calculate the feedback of superconductivity on the magnetic excitations within the spin-exciton scenario (Supplementary Note 1, Supplementary Figs 6–8). The resulting wave vector dependence of the spin-exciton along [0.5, $K$] and [$H$, $H$], which are shown in Fig. 1h,i, respectively, are inconsistent with the experimentally determined upward dispersion (solid lines). Similar dispersive resonances in CeCoIn$_5$ and Ce$_{0.7}$Yb$_{0.3}$CoIn$_5$ (Fig. 3, Supplementary Figs 3 and 4 and Fig. 5) are seen in spite of possible changes in the Fermi surface and superconducting gap symmetry on moving from $x = 0$ to 0.3 (refs 34–36), also inconsistent with the expectation that a spin-exciton should depend sensitively on the Fermi surface. We thus conclude that the upward-dispersing resonance mode in Ce$_{0.95}$Yb$_{0.05}$CoIn$_5$ cannot be interpreted as a spin-exciton arising from the feedback of unconventional $d$-wave superconductivity[12,27,28]. On the other hand, the similarity of the resonance's dispersion along the [$H$, $H$, 0.5] direction with the spin-wave dispersion in AF-ordered nonsuperconducting CeRhIn$_5$ along the same direction[38,39] (Fig. 1j) suggests that the upward-dispersing resonance may be magnon-like. In this scenario, the magnetic resonance arises since the opening of the superconducting gap leads to a strong suppression of Landau damping for preexisting magnon-like excitations, as shown in Fig. 1k,l (Supplementary Note 2 and Supplementary Figs 9–11). This is, therefore, the first experimental observation of a magnetic resonance in an unconventional superconductor that cannot be interpreted as a spin-exciton.

## Results

**Dispersion of the resonance in Ce$_{0.95}$Yb$_{0.05}$CoIn$_5$ along [$H$, $H$, 0.5] and [0.5, 0.5, $L$].** Using a tetragonal unit cell with $a = b = 4.60$ Å and $c = 7.51$ Å for Ce$_{0.95}$Yb$_{0.05}$CoIn$_5$ (Fig. 1a), we define the momentum transfer $\mathbf{Q}$ in three-dimensional reciprocal space in Å$^{-1}$ as $\mathbf{Q} = H\mathbf{a}^* + K\mathbf{b}^* + L\mathbf{c}^*$, where $H$, $K$ and $L$ are Miller indices and $\mathbf{a}^* = \hat{\mathbf{a}}2\pi/a$, $\mathbf{b}^* = \hat{\mathbf{b}}2\pi/b$ and $\mathbf{c}^* = \hat{\mathbf{c}}2\pi/c$. The experiments are carried out using the [$H$, $H$, $L$] and [$H$, $K$, $H$]

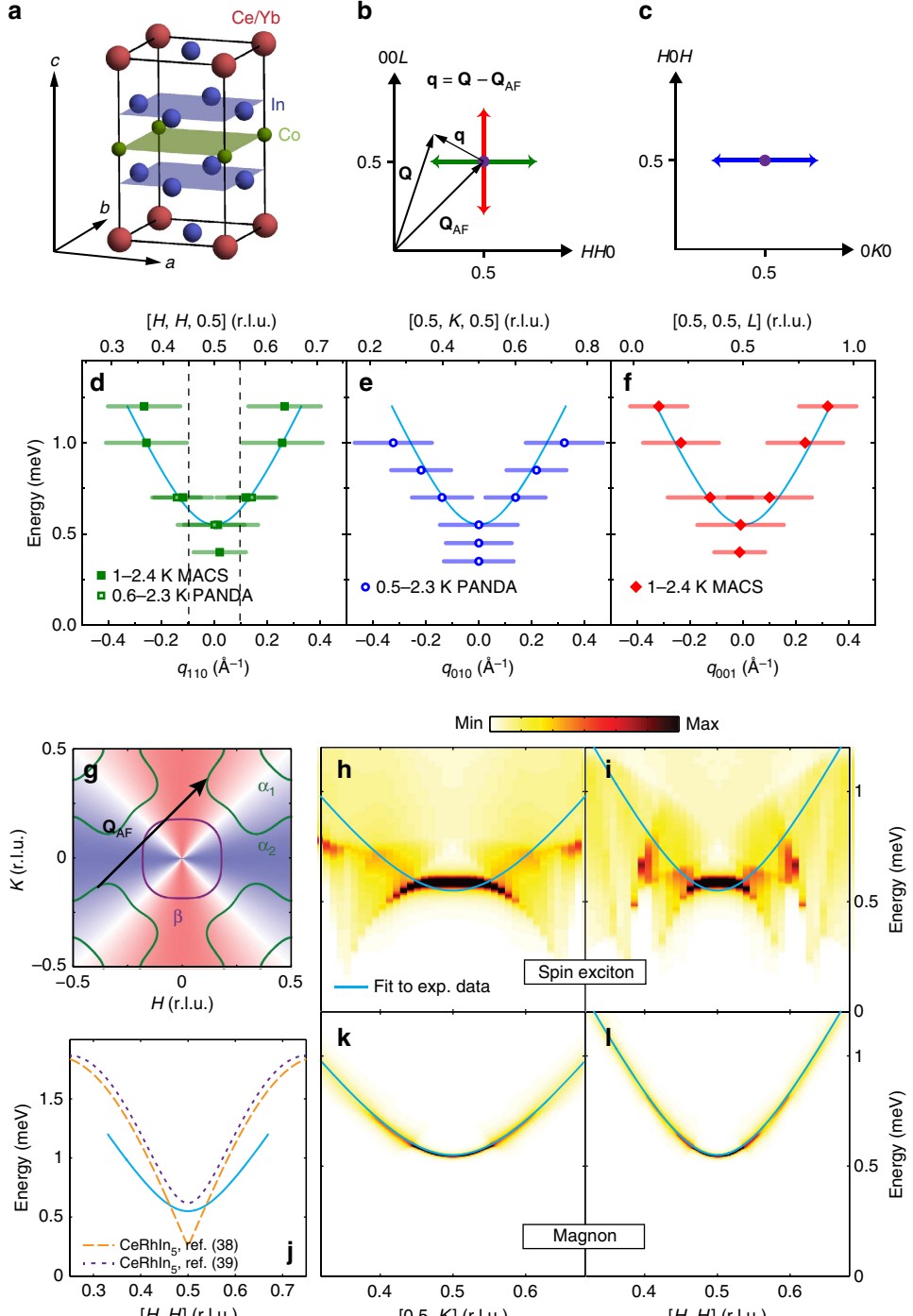

**Figure 1 | Summary of neutron scattering results on $Ce_{0.95}Yb_{0.05}CoIn_5$.** (**a**) Crystal structure of $Ce_{1-x}Yb_xCoIn_5$. (**b**) [H, H, L] scattering plane, where **q** is measured from $\mathbf{Q}_{AF}$ via $\mathbf{q} = \mathbf{Q} - \mathbf{Q}_{AF}$. The red and green arrows represent scans along [0.5, 0.5, L] and [H, H, 0.5] centred at $\mathbf{Q}_{AF}$, respectively. (**c**) [H, K, H] scattering plane. Here scans along [0.5, K, 0.5] centred at $\mathbf{Q}_{AF}$ can be carried out as indicated by the blue arrow. (**d**) Dispersion of the resonance along [H, H, 0.5]. The axis above the figure is **Q** in r.l.u., whereas the axis at the bottom is **q** in Å$^{-1}$. An isotropic dispersion $E = \sqrt{\Delta^2 + (c|\mathbf{q}|)^2}$ ($\Delta = 0.55(1)$ meV, $c = 3.2(1)$ meV·Å) is shown as a cyan solid line, where $\Delta$ represents a spin gap and $c$ is the effective spin wave velocity. The horizontal bars represent experimentally observed peak full-width-at-half-maximum. The dashed vertical lines indicate the ordering wave vector of the so-called Q phase at $\mathbf{Q} = \mathbf{Q}_{AF} \pm (\delta, \delta, 0)$ with $\delta = 0.05$ (ref. 44). (**e,f**) are similar to (**d**), but are for dispersions along [0.5, K, 0.5] and [0.5, 0.5, L], respectively. (**g**) The Fermi surfaces of $CeCoIn_5$, where the blue and red shading represent the d-wave symmetry of the superconductivity order parameter. The black arrow indicates $\mathbf{Q}_{AF}$, which connects parts of Fermi surfaces with sign-reversed superconductivity-order parameters. (**h**) Colour-coded calculated intensity along the [0.5, K] direction by considering the resonance mode to be a spin-exciton. (**i**) Calculated intensity for the spin-exciton along the [H, H] direction. (**j**) Comparison of dispersions of the resonance in $Ce_{0.95}Yb_{0.05}CoIn_5$ (solid cyan line) and spin waves in $CeRhIn_5$ (dashed purple and orange lines)[38,39]. (**k**) Calculated intensity of the resonance along the [0.5, K] direction assuming it is a magnon-like excitation. Dispersion of the magnon-like excitations is obtained from fits to experimental data and the intensity is affected by damping due to the particle−hole continuum. (**l**) Calculated intensity for the magnon-like excitation along the [H, H] direction.

scattering planes to study the dispersions of the resonance along [$H$, $H$, 0.5], [0.5, $K$, 0.5] and [0.5, 0.5, $L$] (Fig. 1b,c). Figure 2a shows the colour-coded plot of the spin excitations at 0.6 K obtained from fits to the raw data at energies $E = 0.3$, 0.55, 0.7, 0.85 and 1 meV along [$H$, $H$, 0.5] for $Ce_{0.95}Yb_{0.05}CoIn_5$ (Fig. 2c). Although the data show a weak commensurate peak at $E = 0.3$ meV, we see a clear commensurate resonance at $E_r \approx 0.55$ meV and upward-dispersing incommensurate peaks for energies $E = 0.7$, 0.85 and 1 meV. Figure 2b shows constant-energy scans at $E = 0.7$ meV below and above $T_c$. At $T = 2.3$ K, we see a broad peak centred at the commensurate AF wave vector $\mathbf{Q}_{AF}$. Upon cooling to below $T_c$ at $T = 0.6$ K, the commensurate peak becomes two incommensurate peaks, which disperse outward with increasing energy (Fig. 2c). Figure 2d shows constant-$\mathbf{Q}$ scans at $\mathbf{Q}_{AF}$ for temperatures $T = 0.6$, 1.5 and 2.3 K. Similar to previous work on pure $CeCoIn_5$ (ref. 12), the data reveal a clear resonance at $E_r \approx 0.55$ meV below $T_c$, and no peak in the normal state above $T_c$.

To further illustrate the dispersive nature of the resonance, we show in Fig. 3 maps of scattering intensities in the [$H$, $H$, $L$] scattering plane of the spin excitations at different energies above and below $T_c$ obtained on the multi-axis crystal spectrometer (MACS) for $Ce_{0.95}Yb_{0.05}CoIn_5$. In the probed reciprocal space, we see clear spin excitations around $\mathbf{Q}_{AF}$, which disperse outward with increasing energy. At an energy ($E = 0.4$ meV) below the resonance, spin excitations are commensurate below (Fig. 3a) and above (Fig. 3b) $T_c$. The constant-energy cuts of the data along the [$H$, $H$, 0.5] direction confirm this conclusion (Fig. 3c). Figure 3d–f shows similar scans at $E = 0.55$ meV and indicate that the scattering becomes broader in reciprocal space. Upon moving to $E = 0.7$ meV (Fig. 3g–i), 1.0 meV (Fig. 3j–l) and 1.2 meV (Fig. 3m–o), we see clear ring-like scattering dispersing away from $\mathbf{Q}_{AF}$ with increasing energy in the superconducting state. The normal-state scattering is commensurate at all energies, and this is most clearly seen in the constant-energy cuts along the [$H$, $H$, 0.5] direction. Based on the difference of data at 2.1 and 1 K in Fig. 3, one can compose the dispersions of the resonance along the [$H$, $H$, 0.5] (Fig. 1d) and [0.5, 0.5, $L$] (Fig. 1f) directions. By plotting the dispersion in Å$^{-1}$ away from $\mathbf{Q}_{AF}$ (q as defined in Fig. 1b), we see that the resonance disperses almost isotropically along these two directions.

**Dispersion of the resonance in $Ce_{0.95}Yb_{0.05}CoIn_5$ along [0.5, $K$, 0.5].** In cuprate superconductors such as $YBa_2Cu_3O_{6.5}$ (ref. 21), $YBa_2Cu_3O_{6.6}$ (ref. 40) and $La_{1.875}Ba_{0.125}CuO_4$ (ref. 41), spin

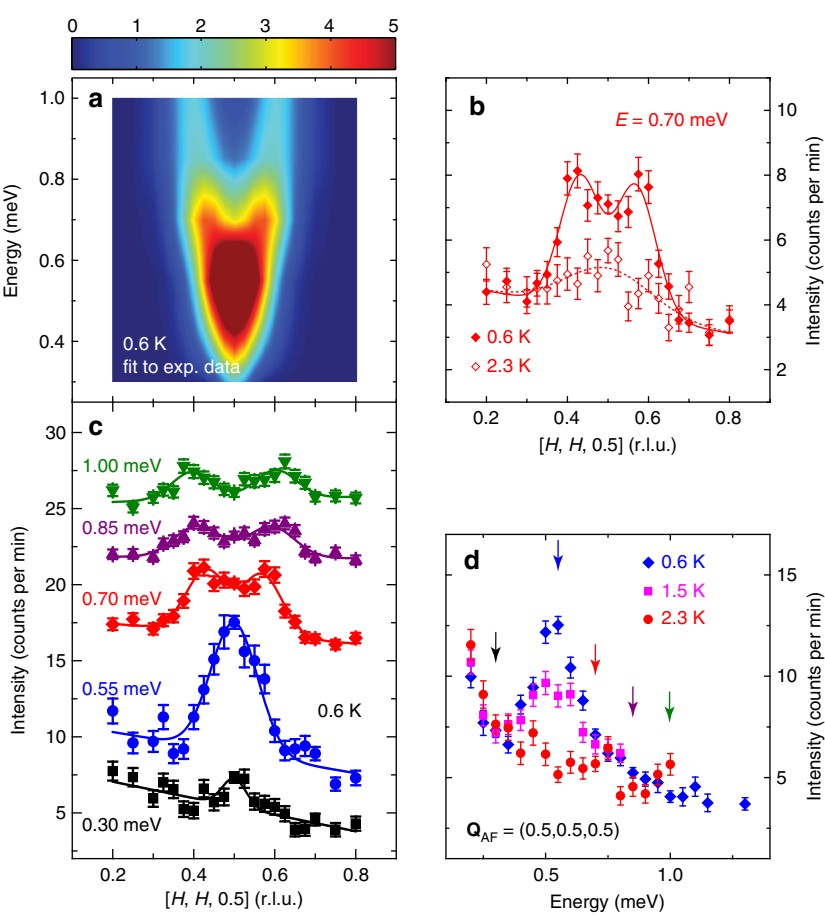

**Figure 2 | Neutron scattering results on $Ce_{0.95}Yb_{0.05}CoIn_5$ in the [$H$, $H$, $L$] scattering plane.** (**a**) Colour-coded intensity of magnetic excitations along [$H$, $H$, 0.5] centred at $\mathbf{Q}_{AF}$ at 0.6 K, obtained from fits to data in (**c**). (**b**) Constant-energy scans along [$H$, $H$, 0.5] centred at $\mathbf{Q}_{AF}$ with $E = 0.7$ meV. The solid symbols are data well below $T_c$ (0.6 K), where two peaks can be resolved whereas open symbols are obtained above $T_c$ (2.3 K) showing a single peak centred at $\mathbf{Q}_{AF}$. The solid line is a fit to the data at 0.6 K with two Gaussian functions, whereas the dashed line is a fit to a single Gaussian function for the data at 2.3 K. Data at the two temperatures are fit simultaneously to have the same linear background. (**c**) Constant-energy scans along [$H$, $H$, 0.5] at 0.6 K. For clarity, scans with $E = 0.55$, 0.75, 0.75 and 1 meV are, respectively, shifted upwards by 5, 13, 18 and 22. The solid lines are fits to either one or two Gaussian functions with a linear background. (**d**) Constant-$\mathbf{Q}$ scans at $\mathbf{Q}_{AF}$. The arrows represent energies for which constant-energy scans are shown in (**c**). All vertical error bars in the figure represent statistical errors of 1 s.d.

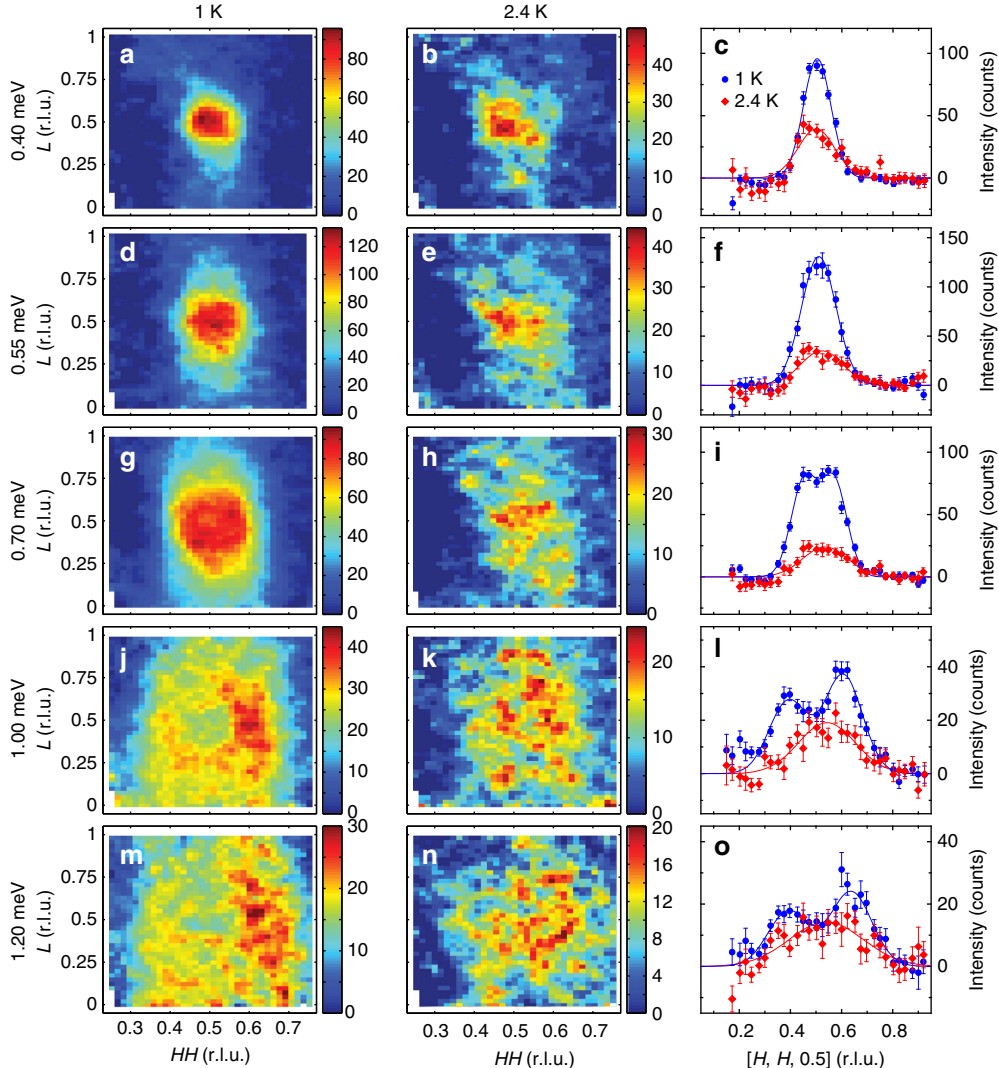

**Figure 3 | Constant-energy maps of scattering intensities in the [H, H, L] scattering plane for Ce$_{0.95}$Yb$_{0.05}$CoIn$_5$.** Constant-energy map at $E = 0.40$ meV at (**a**) 1 K and (**b**) 2.4 K. A |**Q**|-dependent background has been subtracted. (**c**) Cuts obtained from (**a,b**) by binning data with $0.45 \leq L \leq 0.55$; solid lines are fits to the data using either a single or two Gaussian functions. Since a background has already been subtracted in maps in (**a,b**), no background is assumed in the fits. Similarly, (**d–f**) are for $E = 0.55$ meV, (**g–i**) are for $E = 0.70$ meV, (**j–l**) are for $E = 1.00$ meV and (**m–o**) are for $E = 1.20$ meV. All vertical error bars in the figure represent statistical errors of 1 s.d.

excitations above the resonance form a ring-like upward dispersion in the *ab* plane slightly softened from the spin waves in their AF-ordered parent compounds[8]. To conclusively determine if the resonance dispersion is also ring-like in the *ab* plane in Ce$_{0.95}$Yb$_{0.05}$CoIn$_5$, we aligned the single crystals in the $[H, 0, H] \times [0, K, 0]$ ($[H, K, H]$) scattering plane to measure the dispersion of the resonance along $[0.5, K, 0.5]$ centred at $\mathbf{Q}_{AF}$. Figure 4a–f summarizes the constant-energy scans at $E = 0.35$, 0.45, 0.55, 0.7, 0.85 and 1.0 meV along $[0.5, K, 0.5]$. Although the scattering is clearly commensurate at $E = 0.35$ and 0.45 meV below the resonance at $E_r \approx 0.55$ meV (Fig. 4a,b), it becomes incommensurate above the resonance at $E = 0.7$, 0.85 and 1.0 meV with an upward dispersion as a function of increasing energy (Fig. 4d–f). Figure 1e summarizes the dispersion of the resonance in Å$^{-1}$ away from $\mathbf{Q}_{AF}$ along $[0.5, K, 0.5]$. Figure 4g shows the difference of the constant-**Q** scans below and above $T_c$ at $\mathbf{Q}_{AF}$, again revealing a strong peak at the resonance energy of $E_r \approx 0.55$ meV similar to Fig. 2d. Finally, Fig. 4h shows the temperature dependence of the scattering at an incommensurate wave vector (0.5, 0.35, 0.5) and $E = 0.85$ meV,

which reveals a clear superconducting order-parameter-like increase below $T_c$ and indicates that the incommensurate part of the resonance is also coupled to superconductivity.

**Dispersion of the resonance for CeCoIn$_5$ and Ce$_{0.7}$Yb$_{0.3}$CoIn$_5$.** To determine how Yb-doping, and in particular the possible changes in the Fermi surface topology and superconducting gap structure between Yb-doping of $x = 0.1$ and 0.2, affects the behaviour of the resonance[34–36], we carried out additional inelastic neutron scattering experiments on CeCoIn$_5$ and Ce$_{0.7}$Yb$_{0.3}$CoIn$_5$ at MACS. Figure 5a shows temperature differences of constant-**Q** scans at $\mathbf{Q}_{AF}$ below and above $T_c$ in Ce$_{0.7}$Yb$_{0.3}$CoIn$_5$, which reveals a clear resonance at $E_r \approx 0.4$ meV. Figure 5b plots the temperature dependence of the resonance, displaying a superconducting order-parameter-like increase in intensity below $T_c$. From wave vector scans along the $[H, H, 0.5]$ and $[0.5, 0.5, L]$ directions at different energies below and above $T_c$ for Ce$_{0.7}$Yb$_{0.3}$CoIn$_5$ (Supplementary Fig. 5), we can establish the dispersions of the resonance along these two directions as shown in Fig. 5c,d, respectively. Similarly, Fig. 5e,f compares

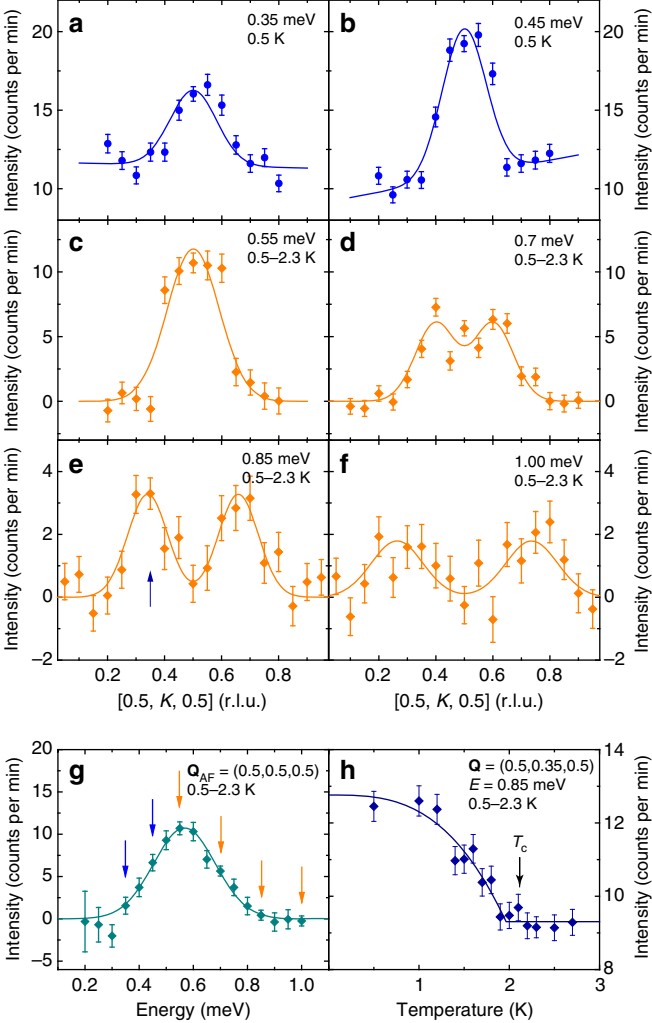

**Figure 4 | Neutron scattering results on Ce$_{0.95}$Yb$_{0.05}$CoIn$_5$ in the [H, K, H] scattering plane.** (**a**) Constant-energy scan along [0.5, K, 0.5] centred at **Q**$_{AF}$ at 0.5 K for E = 0.35 meV. The solid line is a fit to a single Gaussian with a linear background. (**b**) Similar to (**a**), but for E = 0.45 meV. (**c**) Constant-energy scan along [0.5, K, 0.5] centred at **Q**$_{AF}$, obtained by subtracting data at 2.3 K from data at 0.5 K for E = 0.55 meV. The solid line is a fit to a Gaussian function with zero background. (**d**) Similar to (**c**), but for E = 0.7 meV, and the solid line is a fit to two Gaussian functions. (**e**) Similar to (**d**), but for E = 0.85 meV. The arrow points to **Q** = (0.5, 0.35, 0.5), where measurement of the temperature dependence was carried out, shown in (**h**). (**f**) Similar to (**d,e**), but for E = 1.00 meV. (**g**) Constant-**Q** scan at **Q**$_{AF}$ obtained by subtracting the 2.3 K data from the 0.5 K data. The solid line is a Gaussian function centred at E = 0.57(1) meV with zero background. Arrows represent energies at which constant-energy scans are shown in (**a–f**). (**h**) Temperature dependence of scattering intensity at **Q** = (0.5, 0.35, 0.5) for E = 0.85 meV. The solid line is a fit to d-wave superconductivity order parameter with constant background. The superconducting critical temperature $T_c$ obtained from the fit is 2.0(1) K. All vertical error bars in the figure represent statistical errors of 1 s.d.

dispersions of the resonance for CeCoIn$_5$ (Supplementary Fig. 4) and Ce$_{0.95}$Yb$_{0.05}$CoIn$_5$ along the [H, H, 0.5] and [0.5, 0.5, L] directions, respectively. From Fig. 5c–f, we see that the dispersions of the resonance are essentially Yb-doping independent. However, the bottom of the dispersive resonance at **Q**$_{AF}$ moves down in energy with increasing Yb-doping and $E_r$ is proportional to $k_B T_c$, similar to La-doped CeCoIn$_5$ (refs 31,32).

## Discussion

From the dispersions of the resonance along [H, H, 0.5] (Fig. 1d), [0.5, K, 0.5] (Fig. 1e) and [0.5, 0.5, L] (Fig. 1f) for Ce$_{0.95}$Yb$_{0.05}$CoIn$_5$, we see that the mode disperses isotropically in reciprocal space away from **Q**$_{AF}$, which is inconsistent with the resonance being a spin-exciton (see Fig. 1h,i), but resembles a magnon-like excitation with a dispersion similar to spin waves in CeRhIn$_5$ (Fig. 1j, Supplementary Note 3 and Supplementary Fig. 12) that becomes undamped in the superconducting state[16,17]. However, the fact that CeCoIn$_5$ is a multiband system complicates the identification of the resonance's origin. Athough we have assumed here that the main contribution to the resonance arises from the quasi-localized f-levels identified via quasi-particle interference (QPI) spectroscopy in STM experiment[25,28], it is of course possible that there exist further electronic bands that become superconducting and contribute to the resonance (either directly or through a renormalization of the magnetic interaction) but were not detected via QPI spectroscopy. Clearly, further studies are necessary to investigate this possibility.

Moreover, in a recent work on undoped CeCoIn$_5$, it was suggested that the resonance in the energy range of 0.4–0.7 meV is incommensurate along the [H, H, 0.5] direction with wave-vector **Q**$_{AF}$ ± ($\delta$, $\delta$, 0), where $\delta$ = 0.042(2) r.l.u. (ref. 42). Since the incommensurate wave vectors of the resonance appear to be close to the in-plane magnetic field-induced incommensurate static magnetic order at **Q**$_{AF}$ ± ($\delta$, $\delta$, 0) with $\delta$ = 0.05 (the so-called Q phase) (see the vertical dashed lines in Fig. 1d)[43–45], and since it was suggested that the fluctuating moment of the resonance is entirely polarized along the c-axis similar to the ordered moment of the Q phase[12,42], the resonance has been described as a dynamical precursor of the Q phase[46]. Experimentally, we did not observe incommensurate excitations at E = 0.5 meV; nevertheless, our data suggest a smaller splitting than in previous work if the excitations at E = 0.5 are incommensurate (Supplementary Note 4 and Supplementary Fig. 13). Furthermore, the Q phase precursor interpretation of the resonance is also inconsistent with the observed ring-like dispersion at E > 0.7 meV. It is possible that there are more than one contribution to the resonance in CeCoIn$_5$ given its electronic complexity. In the present work, we identify the upward-dispersing magnon-like contribution as being dominant, but do not rule out finer features at lower energies with E < 0.6 meV, which can only be resolved with better resolution. Our data and previous work on CeCoIn$_5$ (ref. 42) are consistent with each other, both showing no signature of a downward dispersion.

Further insight into the nature of the resonance in CeCoIn$_5$ can be gained by considering its behaviour in an applied magnetic field. Previous neutron scattering experiments by Stock et al.[47] observed that the resonance in the superconducting state of CeCoIn$_5$ splits into two modes if a magnetic field is applied along the [1, $\bar{1}$, 0] direction. This splitting into two modes by an in-plane field is rather puzzling, since for a system with a Heisenberg spin symmetry a splitting into three modes is expected. Moreover, if the resonance in CeCoIn$_5$ was entirely polarized along the c-axis[12,42], application of an in-plane magnetic field should not split the resonance into the doublet observed experimentally[47,48]. However, this observation can be explained if the system possesses a magnetic anisotropy with a magnetic easy plane (indicated by the green ellipse in Fig. 6a) that is perpendicular to the direction of the applied magnetic field (red arrow in Fig. 6a). Since the magnetic field applied by Stock et al.[47] lies in the [1, $\bar{1}$, 0] direction, this implies that the easy plane is spanned by the unit vectors in the [0, 0, 1] and [1, 1, 0] directions. This leads us to suggest that the resonance in CeCoIn$_5$ should also have a component along the [1, 1, 0] direction in addition to the c-axis component similar to

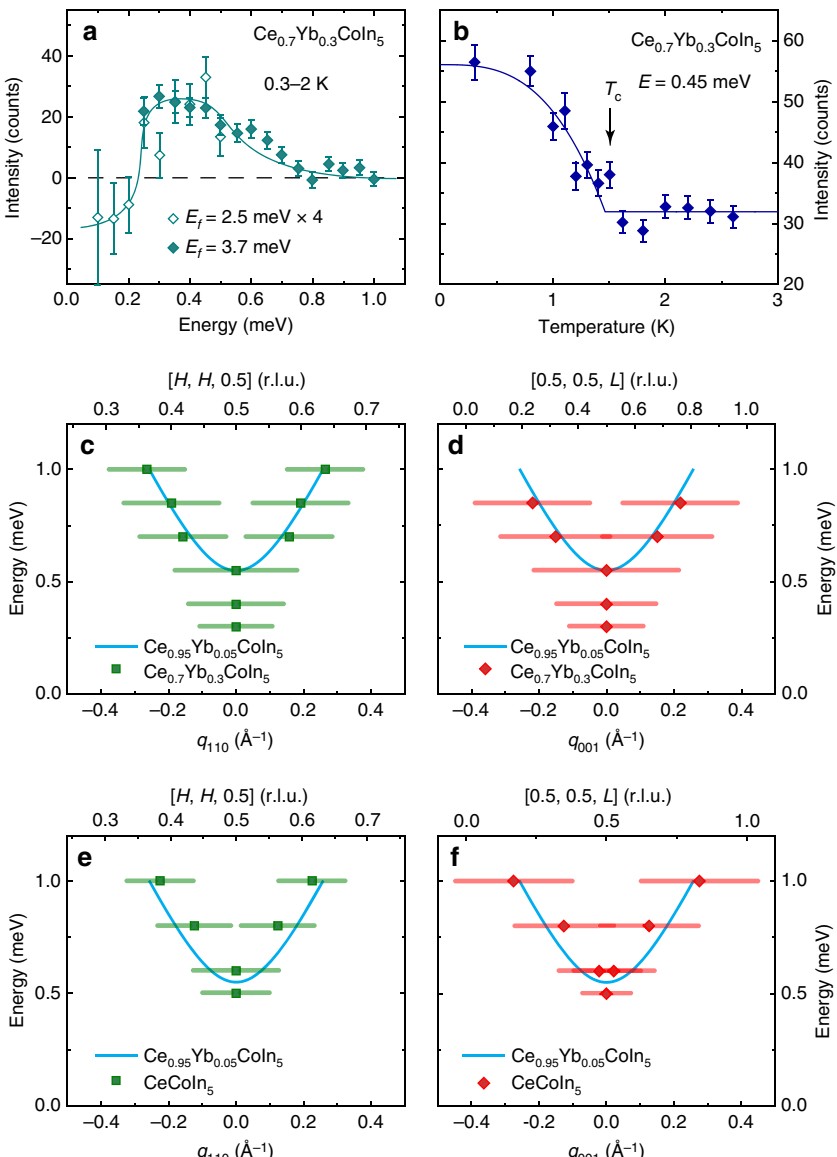

**Figure 5 | Summary of neutron scattering results on Ce$_{0.7}$Yb$_{0.3}$CoIn$_5$ and CeCoIn$_5$.** (a) Difference of constant-**Q** scans at **Q**$_{AF}$ = (0.5, 0.5, 0.5) for 0.3 and 2 K, displaying a resonance mode at $E_r \approx 0.4$ meV for Ce$_{0.7}$Yb$_{0.3}$CoIn$_5$. Filled symbols are obtained with fixed scattered neutron energy $E_f = 3.7$ meV and open symbols are for $E_f = 2.5$ meV scaled up by 4 times. All of the data in the rest of figure are obtained with $E_f = 3.7$ meV. The solid line is a guide to the eye. (b) Temperature dependence of the resonance mode in Ce$_{0.7}$Yb$_{0.3}$CoIn$_5$ for $E = 0.45$ meV and **Q**$_{AF}$ = (0.5, 0.5, 0.5); the solid line is a fit to $d$-wave superconducting gap, with $T_c = 1.5(1)$ K. Dispersion of the resonance along (c) [$H$, $H$, 0.5] and (d) [0.5,0.5,$L$] for Ce$_{0.7}$Yb$_{0.3}$CoIn$_5$. Dispersions of the resonance for CeCoIn$_5$ along [$H$, $H$, 0.5] and [0.5, 0.5, $L$] are showin in (**e,f**), respectively. The solid cyan lines in (**c–f**) are dispersions of the resonance obtained for Ce$_{0.95}$Yb$_{0.05}$CoIn$_5$. The horizontal bars represent experimentally observed peak full-width-at-half-maximum. All vertical error bars in the figure represent statistical errors of 1 s.d.

the resonance in electron-doped iron pnictides[49,50]. Such in-plane spin excitation anisotropy can occur due to the presence of spin-orbit coupling, and does not break the four-fold rotational symmetry of the underlying lattice[50]. The present experimental results do not rule out the presence of such a mode, although it is also challenging to experimentally confirm its presence (Supplementary Note 5 and Supplementary Figs 14 and 15).

To quantitatively understand the effect of a magnetic field on spin excitations, we consider the Hamiltonian (see Supplementary Eq. 1 in ref. 28)

$$H = \sum_{\mathbf{r},\mathbf{r}'} I_{\mathbf{r},\mathbf{r}'} S_{\mathbf{r}} \cdot S_{\mathbf{r}'} + A \sum_{\mathbf{r}} \left(S_{\mathbf{r}}^z\right)^2 - g\mu_B H \sum_{\mathbf{r}} S_{\mathbf{r}}^z \quad (1)$$

with the three terms representing the magnetic interactions

between the $f$-electron moments, the magnetic anisotropy of the system and the interaction with the external magnetic field, respectively. Here, we define the direction of the magnetic field along the [1, $\bar{1}$, 0] direction as the $z$-axis in spin space. We assume $A > 0$, such that the system possesses a hard magnetic axis along [1, $\bar{1}$, 0] and an easy plane (green ellipse in Fig. 6a) perpendicular to it. This Hamiltonian implies that the effective interaction for the longitudinal, non-spin-flip scattering mode (parallel to the applied field) is given by $I_{zz}(\mathbf{q}) = I_{\mathbf{q}} + A$, while the interaction for the transverse mode is given by $I_{\pm}(\mathbf{q}) = I_{\mathbf{q}}$, with $I_{\mathbf{q}}$ being the Fourier transform of $I_{\mathbf{r},\mathbf{r}'}$ in Equation (1). In the vicinity of the AF wave-vector **Q**$_{AF}$, where $I_{\mathbf{Q}_{AF}} < 0$, we thus obtain $|I_{zz}(\mathbf{Q}_{AF})| < |I_{\pm}(\mathbf{Q}_{AF})|$ since $A > 0$ for an easy plane perpendicular to the [1, $\bar{1}$, 0] direction. This implies that the effective interaction

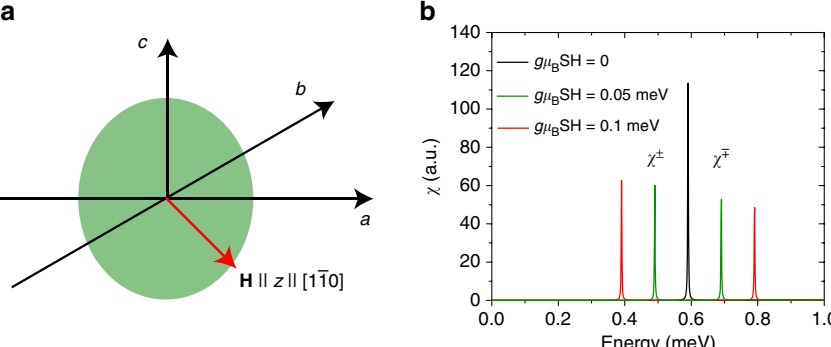

**Figure 6 | Effect of applied magnetic field on the resonance mode. (a)** Orientation of the magnetic field **H** and that of the magnetic easy plane in the crystal lattice. The magnetic field is perpendicular to the magnetic easy plane. **(b)** Evolution of the resonance with increasing magnetic field.

at $\mathbf{Q}_{AF}$ for the longitudinal, non-spin-flip scattering mode (parallel to the applied field) is smaller than for the two transverse, spin-flip scattering modes, which lie in the easy plane. As a result, the longitudinal mode will be located at energies higher than the transverse modes. In particular, for sufficiently large $A$, the longitudinal mode can be located above the onset energy, $\omega_c(\mathbf{Q}_{AF})$, for the particle–hole continuum in the superconducting state, and thus would not emerge as a resonance peak. Hence, only the two transverse modes within the easy plane contribute to the resonance peak. The application of a magnetic field perpendicular to the easy plane of the system then splits the two transverse modes of the resonance peak in energy (while not affecting the longitudinal mode), with the energy splitting increasing linearly with the magnetic field, as shown in Fig. 6b, thus explaining the experimental observation in ref. 47).

If spin excitations in CeCoIn$_5$ are only polarized along the $c$-axis with the existence of an easy axis rather than an easy plane[12,42], with application of a magnetic field along the direction perpendicular to the easy axis along the $[1, \bar{1}, 0]$ direction, the transverse mode along the easy axis shifts down with increasing field, but does not split. Similarly, when a field is applied along the easy axis direction ($c$-axis field), the two transverse modes are located at higher energies, while the longitudinal mode, which is located at lower energies, does not split in the magnetic field. The presence of a longitudinal spin excitation along the $[1, 1, 0]$ direction is also consistent with the magnetic field effect work of ref. 48, where the resonance is believed to be a composite excitation, which contains three excitation channels involving both transverse and longitudinal modes.

While unconventional superconductivity in copper oxide, iron pnictide and heavy fermion superconductors appears with the suppression of the static AF order in their parent compounds, dispersive magnon-like excitations persist in the doped superconductors[8,10,51]. Our discovery that the resonance itself in Ce$_{1-x}$Yb$_x$CoIn$_5$ shows a robust ring-like upwards dispersion suggests that, instead of being a spin-exciton in a $d$-wave superconductor[2,7], the resonance may be a magnon-like excitation revealed in the superconducting state[17]. Since the presence of a propagating spin resonance is characteristic of a nearby AF state, we propose that the magnon-like resonance mode in Ce$_{1-x}$Yb$_x$CoIn$_5$ is the strong-coupling analogue of a weak coupling spin-exciton. This would imply that the nature of the magnetic resonance—spin-exciton versus magnon-like excitation—represents a new criterion to distinguish between more weakly and more strongly coupled unconventional superconductors.

## Methods

**Sample preparation.** Single crystals of Ce$_{1-x}$Yb$_x$CoIn$_5$ ($x = 0$, 0.05 and 0.3) were prepared by the indium self-flux method. Details of sample preparation and characterizations have been previously reported; lattice parameters for Ce$_{1-x}$Yb$_x$CoIn$_5$ remain similar to pure CeCoIn$_5$ for all reported doping levels[33]. We use the nominal doping throughout the paper to be consistent with earlier work[33], although the actual doping is ~1/3 of the nominal doping[52]. Supplementary Fig. 1a shows the out-of-phase AC magnetic susceptibility (15.9 Hz) measured on Ce$_{1-x}$Yb$_x$CoIn$_5$ samples with $x = 0.05$ and 0.3 from the same growth batches used for neutron scattering experiments. Bulk superconductivity appears at $T_c = 2.25$ K and $T_c = 1.5$ K, respectively, whereas $T_c = 2.3$ K in pure CeCoIn$_5$ (ref. 33).

Hundreds of Ce$_{1-x}$Yb$_x$CoIn$_5$ single crystals with total masses of 0.8, 2.5 and 1.4 g, respectively, for $x = 0$, 0.05 and 0.3 were co-aligned on several aluminium plates using CYTOP as hydrogen-free glue (Supplementary Fig. 1b). The plates are then mounted in either the $[H, H, 0] \times [0, 0, L]$ ($[H, H, L]$) (Supplementary Fig. 1c) or the $[H, 0, H] \times [0, K, H]$ ($[H, K, H]$) scattering plane (Supplementary Fig. 1d). The total thickness of samples on co-aligned plates is 1–2 mm, minimizing neutron absorption due to indium. Absorption becomes most significant when the incident or the scattered neutron beam becomes perpendicular to $[0, 0, 1]$, which does not occur for reciprocal space regions shown in this work.

**Experiment details and analysis.** Neutron scattering experiments were carried out on the PANDA cold triple-axes spectrometer[53] at Heinz Maier-Leibnitz Zentrum and the MACS instrument at the NIST Center for Neutron Research. The experiments on PANDA used a Be filter 180 mm in length after the sample, which is highly effective in removing contamination from higher-order neutrons; both the analyser and the monochromator are doubly focused to maximize neutron flux at the sample. Vertical focusing of the analyser is fixed, whereas horizontal focusing is variable. Both the horizontal and vertical focusing of the monochromator are variable. The variable focusings are adjusted depending on the neutron wavelength, which is based on empirically optimized values. The PANDA experiment in the $[H, H, L]$ scattering plane used a fixed $k_f$ of 1.3 Å$^{-1}$ ($E_f \approx 3.5$ meV) and the experiment in the $[H, K, H]$ scattering plane used a fixed $k_f$ of 1.57 Å$^{-1}$ ($E_f \approx 5.1$ meV). The MACS experiments in the $[H, H, L]$ scattering plane used Be filters both before and after the sample with fixed $E_f = 3.7$ meV. MACS consists of 20 spectroscopic detectors, each separated by 8°. By rotating the sample and shifting all of the detectors to bridge the 8° gaps, a map in terms of sample rotation angle and scattering angle at a fixed energy transfer can be efficiently constructed. A significant portion of the reciprocal space in the scattering plane can be covered, which further allows cuts along the high-symmetry directions. Ninety-degree collimators are used between the sample and each individual analysers. The analysers are vertically focused, while the monochromator is doubly focused.

For the neutron scattering results on PANDA, a linear background is assumed for all measured constant-energy scans, while no background is used for scans obtained by subtracting data above $T_c$ from those obtained below $T_c$. The constant-energy scans are then simply fit to either one or two Gaussian peaks. For the neutron scattering results obtained on MACS, maps of large portions of the scattering plane for several energy transfers were collected both below and above $T_c$. A |$\mathbf{Q}$|-dependent background is obtained by masking the signal near (0.5,0.5,0.5) and is then fit to a polynomial. The signal with |$\mathbf{Q}$| < 0.5 Å$^{-1}$ is masked throughout the analysis. The fit background is then subtracted from the map and the data are folded into the first quadrant of the scattering plane to improve statistics. The results for Ce$_{0.95}$Yb$_{0.05}$CoIn$_5$ are shown in Fig. 3 and Supplementary Fig. 3. Cuts along $[H, H, 0.5]$ are obtained by binning data with $0.45 \leq L \leq 0.55$ and fit with a single or two Gaussian peaks. Cuts along $[0.5, 0.5, L]$ are obtained by binning data with $0.45 \leq H \leq 0.55$ and fit by a sum of

Lorentzian peaks, accounting for the $Ce^{3+}$ magnetic form factor $f(\mathbf{Q})$ and the polarization factor assuming excitations are dominantly polarized along the $c$-axis similar to previous work[12]. The possible presence of excitations polarized along the [1, 1, 0] direction is discussed in Supplementary Note 5. The function used to fit scans along [0.5, 0.5, L] can be written as

$$I(\mathbf{Q}) \propto f(\mathbf{Q})^2 \left(1 - (\hat{\mathbf{Q}} \cdot \hat{\mathbf{c}})^2\right) \sum_{n=-\infty}^{\infty} F(n+L) \qquad (2)$$

where $F(L)$ is either a single Lorentzian peak centred at $L = 0.5$ or two Lorentzian peaks equally displaced from $L = 0.5$. The peaks along [0.5, 0.5, L] are significantly broader compared to those along [H, H, 0.5], and remain non-zero even for $L = 0$ (Supplementary Fig. 3). This contrasts with similar scans along [H, H, 0.5] in Fig. 3, where the intensity drops to zero away from $\mathbf{Q}_{AF}$. MACS data of $CeCoIn_5$ and $Ce_{0.7}Yb_{0.3}CoIn_5$ with the corresponding maps and cuts are shown in Supplementary Figs 4 and 5. Similar to $Ce_{0.95}Yb_{0.05}CoIn_5$, the resonance mode clearly disperses upward with increasing energy.

**Data availability**. The data that support the findings of this study are available from the corresponding author upon request.

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

## Acknowledgements

We thank Qimiao Si, S. Raymond and C. Stock for helpful discussions. We also thank S. Raymond for sharing with us his unpublished data on $CeCoIn_5$. We acknowledge help from Mengshu Liu, Xingye Lu and Wenliang Zhang for assistance with sample co-alignment, and Scott Carr, Weiyi Wang and Jose Rodriguez for preliminary measurements on $Ce_{0.7}Yb_{0.3}CoIn_5$. The neutron scattering work at Rice is supported by the U.S. DOE, BES, under Grant No. DE-SC0012311 (P.D.). Part of the material characterization efforts at Rice is supported by the Robert A. Welch Foundation Grant No. C-1839 (P.D.). The research at UCSD was supported by the U.S. DOE, BES, under Grant No. DE-FG02-04ER46105 (sample synthesis), and the U.S. NSF, under Grant No. DMR-1206553 (sample characterization). The work by JVD and DKM was supported by the U.S. DOE, BES, under Grant No. DE-FG02-05ER46225. The research at National Institute of

Standards and Technology is in part supported by U.S. NSF, under Agreement No. DMR-1508249. The research at Fudan University is in part supported by the NSFC, under Grant No. 11474060.

## Author contributions

The samples were prepared by I.K.L., B.D.W., S.J., D.Y., L.S. and M.B.M. Neutron scattering experiments were carried out by Y.S., A.S., P.C., Y.Q., and P.D. Data analysis was done by Y.S. Theoretical calculations were done by J.V. and D.K.M. The paper was written by P.D., D.K.M., and Y.S. with input from all co-authors.

## Additional information

**Competing financial interests:** The authors declare no competing financial interests.

