## [Peer review file · Nature Communications]

Reviewers' Comments:

Reviewer #1 (Remarks to the Author)

Having read the previous correspondence of the authors with the referees, the revised article, and the extensive supplementary information file, I conclude that the authors made an impressive effort to answer all the referees' questions and to provide additional evidence supporting their interpretation. Some of the most important points of criticism were raised by more than one referee, in particular the importance of discussing the field-induced splitting of the resonant mode. This point has been addressed in full in the authors' reply, and I especially appreciate that the authors got in contact with other groups working on the same material in an attempt to come to a consensus and explain their results in the context of the numerous available studies. I also appreciated that the authors included a comparison of their data with the parent compound, following my recommendation. In my opinion, the new version of the paper is a significant improvement over the original submission, and it meets the publication criteria of Nature Communications. My only recommendation is that the authors consider including at least some of their Supplementary Materials into the main text. As Nature Communications doesn't have a strict page limit, and allows for an unlimited Methods section that is currently missing in the paper, most of the Supplementary Materials could be incorporated in the main text and further improve the article. In particular, the description of the sample preparation and other technical aspects could be described in the Methods, and the theoretical part related to the RPA calculations could be included as part of the main text.

Reviewer #2 (Remarks to the Author)

The paper describes the resonance peak in Yb doped CeCoIn₅ and in particular highlights the presence of an upward dispersion in this compound. The results are interpreted in terms of a paramagnon and statements are made regarding the polarization of the fluctuations along with comparisons to previous field dependence.

While the results are of high quality, I still believe that the interpretation in terms of a paramagnon is not substantiated by the data. Given that this is the central point of the paper, I am not able to recommend it for publication. I list my criticisms of the new manuscript below.

1) Incommensurate resonance:

Looking at the resolutions of the spectrometers, the authors' claim that the resonance is commensurate is not experimentally realistic. The authors of PRL 115, 037001 set up the spectrometer to have very good momentum resolution (see the horizontal bar in Fig 2). The configurations employed on both Panda and MACS, in my view, cannot match this. On this point, if the authors are going to make this conclusion, they should at least state the momentum resolution, compare it with previous studies which have found an incommensurate resonance, and how it was determined. The commensurate (in momentum) seems to be important for the interpretation in terms of paramagnons so this needs to be addressed appropriately.

2) Fluctuations parallel to [110]:

The idea of the longitudinal fluctuations dominating the resonance is very interesting and based on the elegant work in PRL 111, 107006. However, I don't entirely see the analogy. The pnictides/chalcogenide superconductors are all proximate to a tetragonal-orthorhombic distortion which breaks the four fold symmetry. There is no such evidence in any 115 system that there is such symmetry breaking (that I am aware of). Are there any electronic properties measurements that substantiate this claim? If there is no evidence of such symmetry breaking I think the suggestion is highly speculative and not appropriate for publication. I think the neutron measurements previously reported were sensitive to fluctuations along [1-10].

3) Magnetic field results:

The authors have gone through considerable length to try and understand the magnetic field splitting of the resonance reported previously. While this is commendable, I have some questions regarding the discussion and I found many of the conclusions very hard to follow. I don't understand why the authors are trying to force this into a supplementary information component and not write a longer paper on this that has all of the details. This is of course up to the authors, but I just state this as I have concerns how appropriate this is for supplementary information in Nature Communications.

There is no magnetic Bragg peak so how relevant is the spin-wave analysis? What is the model prediction for a field along c , or when the field is located in the easy-plane? I got the impression that the supplementary information states there is no effect while PRL 109, 167207 (2012) shows a strong suppression of the resonance with a 3.5 T field aligned along this direction (Fig 1 d). Is this consistent with results from the authors?

I do not fully understand the authors statement about crystal fields. In the presence of a molecular field (and hence $J(Q)$), one can get dispersion to the lowest energy (ground state) doublet in Ce based compounds. These are the low-energy spin waves the authors refer to. I think there is a serious semantics problem here and the authors' point needs to be clarified.

4) Similarity with CeRhIn5?:

I realize this was asked in a previous report, but I am still unclear on how the dispersion compares with CeRhIn5. There were two types of modes found in that system with one polarized out of the a - b plane and one within the a - b plane. Which one is plotted? If the authors want to give the full picture the data should be compared against both modes. Looking at the results, it seems the dispersion is more comparable with the modes polarized out of the a - b plane in CeRhIn5.

5) Dispersion of resonance in comparison to cuprates?:

The upward dispersion measured looks a lot like what has been measured in a number of cuprates. I am therefore puzzled by the comparison with theory in the reply to previous comments. I also find the fit in Figure 2 a) somewhat misleading. Are there actually any measurements presented below the resonance energy here showing no downward dispersion. I don't believe the authors have good enough resolution to really give a definitive measurement below the resonance energy. I might be wrong on this, I just don't think this point is entirely made from the available data.

6) Spectral weight?

How do the absolute spectral weights compare between Yb doped and pure compounds? If comparisons are going to be drawn resulting in strong conclusions, this point needs to be addressed. This might be hard to address, but is there an analogy with CeRhIn5 - are the spectral weights similar?

7) Absorption

The neutron absorption in an extreme case has been modelled and compared against data in PRL 114, 237005 (see supplementary information). Is the author's analysis consistent with this? Note that at large values of L the absorption becomes less of a problem which seems to be inconsistent with this current paper. I don't understand why the analysis worked for CeRhIn5 and not in CeCoIn5 despite the former being a stronger absorber. This brings the L dependence claimed by the authors into question.

In summary, while the data is of high quality, the strong conclusions made in this paper are not substantiated by the data. The results showing an upward dispersion are new and there is likely comparisons to be made with the cuprates and pnictides like the authors have attempted. They have also done substantial work in understanding the field dependence which is commendable. However, due to the central interpretation of paramagnon scattering and the many unsubstantiated statements (list above) along with a poor comparison with previous works, I cannot recommend the paper for publication in its current form.

Reviewer #3 (Remarks to the Author)

The authors have made several revisions from the previous version of the manuscript. The quality of the data remains high, and the addition of the new CeCoIn5 results has strengthened the experimental aspects of the work. Unfortunately my previous concerns about the interpretation are not really satisfied by the authors' response or by the revisions.

The authors continue to insist on two distinct scenarios for the resonance excitation: either excitons as found in weak coupling band + RPA models, or propagating spin-wave-like modes characteristic of a cooperative paramagnet --- black or white. The authors clearly show that their results are not consistent with a pure spin exciton as obtained from one particular weak coupling theory, and I accept this. However, the electronic system is very complex, with likely several different f- and d-orbitals contributing to the magnetic response, and with the added complication of a quantum coherent superconducting state. With all this in the mix I feel it is over-simplistic to conclude that because there is an upwards curvature the spin resonance must be a paramagnon. There may well be some paramagnon character, but given the nature of the system, the susceptibility that is measured in the neutron spectra could derive from several different electronic contributions which cannot be separated in the experiment.

I would like to see the experimental results published because, as I said before, it is a very thorough study which makes a significant contribution to the field of unconventional superconductivity. I cannot recommend publication with the current strong conclusion unequivocally identifying the resonance as a paramagnon, but I think that if the authors were to considerably soften this conclusion while retaining the falsification of the spin excitation scenario then the paper would be suitable for publication.

Reviewers' comments:

Reviewer #1 (Remarks to the Author):

In my opinion, the authors have addressed all the criticism raised by all referees, and I recommend the paper for publication in its present form.

Reviewer #2 (Remarks to the Author):

The paper is a resubmission based on my previous comments. The authors have softened some of their claims, but I still cannot recommend the paper for publication based on the points raised previously by myself and the other referees.

The authors still insist on heavily pushing the magnon picture with now the emphasis on their being the lack of any apparent downward dispersion. The authors don't have the resolution to make this claim and the results from Raymond et al are not conclusive either way. If there is no data, the authors cannot use this as a basis as stated in the abstract. I also find the new discussion regarding the field dependence highly speculative (although very interesting I have to admit).

In summary, I still cannot recommend the paper for publication. The paper is heavily biased and not substantiated by data.

Reviewer #3 (Remarks to the Author):

The authors have slightly softened their previous strong conclusion that the neutron spectrum supports a magnon interpretation. However, the manuscript still doesn't contain any discussion of the complexity of the magnetic response. In my view the revisions are still not adequate, and so regrettably I cannot recommend publication.

REVIEWERS' COMMENTS:

Reviewer #3 (Remarks to the Author):

The authors have made a serious attempt to address my concerns and those of the other referees. In particular, the explanation of the magnetic dispersion in the vicinity of the resonant mode is more balanced now. There is also a good discussion of possible magnetic field effects. I can now recommend the manuscript for publication.

Reviewers' comments:

Reviewer #1 (Remarks to the Author):

Having read the previous correspondence of the authors with the referees, the revised article, and the extensive supplementary information file, I conclude that the authors made an impressive effort to answer all the referees' questions and to provide additional evidence supporting their interpretation. Some of the most important points of criticism were raised by more than one referee, in particular the importance of discussing the field-induced splitting of the resonant mode. This point has been addressed in full in the authors' reply, and I especially appreciate that the authors got in contact with other groups working on the same material in an attempt to come to a consensus and explain their results in the context of the numerous available studies. I also appreciated that the authors included a comparison of their data with the parent compound, following my recommendation. In my opinion, the new version of the paper is a significant improvement over the original submission, and it meets the publication criteria of Nature Communications. My only recommendation is that the authors consider including at least some of their Supplementary Materials into the main text. As Nature Communications doesn't have a strict page limit, and allows for an unlimited Methods section that is currently missing in the paper, most of the Supplementary Materials could be incorporated in the main text and further improve the article. In particular, the description of the sample preparation and other technical aspects could be described in the Methods, and the theoretical part related to the RPA calculations could be included as part of the main text.

We appreciate the positive comments from the referee, following the referee's suggestion we have moved most of the sample preparation and technical details to the 'Methods' section, and moved substantial amount of theoretical treatment of the resonance to the main text. However, we decided not to move the RPA calculation to the main text as this has been well-documented many times.

Reviewer #2 (Remarks to the Author):

The paper describes the resonance peak in Yb doped CeCoIn₅ and in particular highlights the presence of an upward dispersion in this compound. The results are interpreted in terms of a paramagnon and statements are made regarding the polarization of the fluctuations along with comparisons to previous field dependence.

While the results are of high quality, I still believe that the interpretation in terms of a paramagnon is not substantiated by the data. Given that this the central point of the paper, I am not able to recommend it for publication. I list my criticisms of the new manuscript below.

We acknowledge the comments that our data are of high quality, and answer the referee's criticisms below.

1) Incommensurate resonance:

Looking at the resolutions of the spectrometers, the authors' claim that the resonance is commensurate is not experimentally realistic. The authors of PRL 115, 037001 set up the spectrometer to have very good momentum resolution (see the horizontal bar in Fig 2). The configurations employed on both Panda and MACS, in my view, cannot match this. On this point, if the authors are going to make this conclusion, they should at least state the momentum resolution, compare it with previous studies which have found an incommensurate resonance, and how it was determined. The commensurate (in momentum) seems to be important for the interpretation in terms of paramagnons so this needs to be addressed appropriately.

We agree that our resolution is not as good as that of PRL 115, 037001. But as we have shown in our measurement at $E=0.5\text{meV}$ for pure CeCoIn_5 , incommensurate spin excitations, if present at all, must have a splitting that is smaller than $\delta=0.042$ as suggested previously (PRL 115, 037001. This is made more clear in Supplementary Note 1). Further, there is no evidence of incommensurate spin excitations at energies below the resonance, say, at $E=0.4\text{meV}$, regardless of the instrument resolution. To confirm this is also the case in PRL 115, 037001, we contacted the authors of the paper and requested the raw data from their work. Enclosed please find our e-mail communication with Dr. Stephane Raymond and the raw data from his work. Even in his case, there is NO evidence of incommensurate spin excitations (as agreed by him in the private e-mail). Therefore, our data and Raymond's raw data are fully consistent with each other, and the difference lies solely in his and our interpretation of the data. We have made this clear in the revised draft, clearly indicating that the raw data in our work is consistent with those of PRL 115, 037001.

2) Fluctuations parallel to [110]:

The idea of the longitudinal fluctuations dominating the resonance is very interesting and based on the elegant work in PRL 111, 107006. However, I don't entirely see the analogy. The pnictides/chalcogenide superconductors are all proximate to a tetragonal-orthorhombic distortion which breaks the four fold symmetry. There is no such evidence in any 115 system that there is such symmetry breaking (that I am aware of). Are there any electronic properties measurements that substantiate this claim? If there is no evidence of such symmetry breaking I think the suggestion is highly speculative and not appropriate for publication. I think the neutron measurements previously reported were sensitive to fluctuations along [1-10].

We are proposing that the excitations polarized along the in-plane longitudinal direction are different from those polarized along the in-plane transverse direction. For $Q=(0.5,0,5,0.5)$, these two directions would correspond to the [1,1,0] and [1,-1,0] directions. A difference in the

polarization of the magnetic excitations in these two directions does not break four fold symmetry of the system (see discussion of Fig.7b in the SI). The case is the same in PRL 111, 107006, where such an anisotropy exists also in the tetragonal state and also does not break four fold symmetry of the system. The distinction being between in-plane longitudinal and transverse directions that hence does not break four-fold symmetry is now emphasized in main text and Supplementary Note 4. Whether CeCoIn5 or the iron pnictides display a tetragonal to orthorhombic structural transition is irrelevant for this observation and conclusion.

3) Magnetic field results:

The authors have gone through considerable length to try and understand the magnetic field splitting of the resonance reported previously. While this is commendable, I have some questions regarding the discussion and I found many of the conclusions very hard to follow. I don't understand why the authors are trying to force this into a supplementary information component and not write a longer paper on this that has all of the details. This is of course up to the authors, but I just state this as I have concerns how appropriate this is for supplementary information in Nature Communications.

Following the suggestion of the referee, we are now writing a separate paper on the theoretical treatment of the splitting of the resonance under applied field in the presence of anisotropies, and have therefore moved much of the original supplementary information into the 'Methods' section and main text. Since understanding the magnetic field effect is a major part of the work, we outline the key results of our theoretical study in the main text.

There is no magnetic Bragg peak so how relevant is the spin-wave analysis? What is the model prediction for a field along c , or when the field is located in the easy-plane? I got the impression that the supplementary information states there is no effect while PRL 109, 167207 (2012) shows a strong suppression of the resonance with a 3.5 T field aligned along this direction (Fig 1 d). Is this consistent with results from the authors?

The absence of magnetic Bragg peaks does not exclude magnon-like excitations, as is well known in the cuprates and iron pnictides.

In PRL 109, 167207 (2012) a field of 3.5T was applied along c -axis at 2 K and weak magnetic excitations were seen, whereas at 0T a resonance can be observed. While this is interesting, we note that at 2K, a 3.5 T c -axis aligned field can in fact drive CeCoIn5 into the normal state (see <http://arxiv.org/pdf/1103.0564v4.pdf>, New J Phys 13, 113039, $T_c \sim 1.6K$ for 3.5T along c -axis), or at best barely in the superconducting state, whereas at 0 T, 2K is considerably below $T_c = 2.3 K$. Therefore, it is expected that there would be no resonance with 3.5T applied along c at 2K. It is currently not known how the resonance behaves with a c -axis magnetic field experimentally.

I do not fully understand the authors statement about crystal fields. In the presence of a molecular field (and hence $J(Q)$), one can get dispersion to the lowest energy (ground state) doublet in Ce based compounds. These are the low-energy spin waves the authors refer to. I think there is a serious semantics problem here and the authors' point needs to be clarified.

We agree with the referee that the dispersive features are due to exchange interactions $J(Q)$. We now emphasize that the upward dispersion is inconsistent with the spin-exciton picture, with the magnon scenario being a possible explanation.

The crystal field interpretation presented in PRL 109, 167207 (2012) comes from the following observations: (1) the splitting of the modes is $E = \pm 1/2 g H \mu_B$ and (2) the spectral weight of the mode is consistent with the matrix elements expected for intra-doublet transition.

Regarding (1), PRL 109, 167207 (2012) does not explain why an apparent $S=1/2$ is attributed to the ground state doublet. From PRB 70, 134505, the ground state eigenfunctions are $\alpha|\pm 5/2\rangle - \beta|\mp 3/2\rangle$, clearly not $S=1/2$. It is also not clear how Lande factor g was obtained. Without these details it is not possible to determine validity of the suggestion that the splitting of the resonance is consistent with splitting of the ground state doublet.

Regarding (2) the spectral weight is obtained assuming the resonance is Q -independent and only integration over energy is performed (ideally an average over Q should also be performed). This is clearly an assumption that could lead to gross overestimated spectral weight, since the resonance has strong Q -dependence as can be seen in PRL 100, 087001 (2008).

Given both evidence used to argue for the resonance being a CEF excitation are questionable, we do not find the CEF scenario appealing.

4) Similarity with CeRhIn5?:

I realize this was asked in a previous report, but I am still unclear on how the dispersion compares with CeRhIn5. There were two types of modes found in that system with one polarized out of the a - b plane and one within the a - b plane. Which one is plotted? If the authors want to give the full picture the data should be compared against both modes. Looking at the results, it seems the dispersion is more comparable with the modes polarized out of the a - b plane in CeRhIn5.

We plotted the one that is polarized in-plane for CeRhIn5. As we have already clarified in the previous version of the manuscript, a comparison between CeRhIn5 and CeCoIn5 is shown to demonstrate that the upward dispersion is due to a similar nearest-neighbor J and no comparison is made in terms of polarizations.

5) Dispersion of resonance in comparison to cuprates?:

The upward dispersion measured looks a lot like what has been measured in a number of cuprates.

I am therefore puzzled by the comparison with theory in the reply to previous comments. I also find the fit in Figure 2 a) somewhat misleading. Are there actually any measurements presented below the resonance energy here showing no downward dispersion. I don't believe the authors have good enough resolution to really give a definitive measurement below the resonance energy. I might be wrong on this, I just don't think this point is entirely made from the available data.

While an hourglass dispersion is almost universally observed in the cuprates, the resonance mode, defined as enhancement of magnetic excitations due to superconductivity, shows a prominent downward dispersion (see Science 288, 1234 (2000) and PRL 93, 207003 (2004)). Even though an upward dispersing mode was suggested experimentally in PRL 93, 207003 (2004)), this branch is much weaker than the downward dispersing branch, and is not directly connected to the resonance at (π, π) (it is therefore referred to as the Q^* mode). This mode is not part of the upper part of the hour glass dispersion. More importantly, the Q^* mode can be understood within RPA spin-exciton model (see PRL 94, 147001 (2005)), and we see the same feature in our current RPA calculation for CeCoIn5 (the weak side features above the downward dispersing resonance in Fig 1j and 1i). Furthermore, this feature is anisotropic along $[H, H, 0]$ and $[0, K, 0]$ directions, in contrast to the isotropic dispersion seen experimentally in CeCoIn5. These differences mark the key findings of our paper, while the spin-exciton theories predict the same results for CeCoIn5 and the cuprates, the experiments show completely different behavior.

From our data at $E=0.3\text{meV}$ and the data at $E=0.4\text{meV}$ that we received from the authors of PRL 111, 107006 (through private communications), no downward dispersion can be resolved. While there may nonetheless be a downward dispersion that is currently not observable, the upward dispersion we uncover is clearly the most prominent feature, in stark contrast to the cuprates and the spin-exciton predictions.

6) Spectral weight?

How do the absolute spectral weights compare between Yb doped and pure compounds? If comparisons are going to be drawn resulting in strong conclusions, this point needs to be addressed. This might be hard to address, but is there an analogy with CeRhIn5 - are the spectral weights similar?

The highly neutron absorbing nature of In in CeCoIn5 and Yb-doped materials makes this comparison difficult. We have now softened our conclusion. Moreover, in the case of cuprates the magnetic spectral weight clearly decreases with doping whereas in the iron pnictides the spectral weight does not change much with doping, although both systems exhibit magnon like excitations. Therefore, it is not clear how knowing the change of spectral weight with Yb doping can shed light on whether the resonance is a magnon-like excitation.

7) Absorption

The neutron absorption in an extreme case has been modelled and compared against data in PRL 114, 237005 (see supplementary information). Is the author's analysis consistent with this? Note that at large values of L the absorption becomes less of a problem which seems to be inconsistent with this current paper. I don't understand why the analysis worked for CeRhIn5 and not in CeCoIn5 despite the former being a stronger absorber. This brings the L dependence claimed by the authors into question.

The absorption calculation in PRL 114, 237005 uses a different E_f and E compared to our measurement. We are unable to comment on the accuracy of the absorption correction in PRL 114, 237005, but for our E_f and E absorption is definitely stronger at $L=1.5$ than 0.5 . This fact can be readily seen in Fig. 2 of PRL 100, 087001 (2008), where the intensity at $L=1.5$ is strongly suppressed, even when the excitations were assumed to be purely c-axis polarized.

We have simply plotted the scattering triangles with respect to the sample to explain why the absorption will be stronger at $L=1.5$, when k_i or k_f is close to $[110]$, absorption will be strong. This is now made more clear in Supplementary Note 4. On this point, we have also consulted Dr. Stephane Raymond, and he also agrees with us that absorption is much stronger at $L=1.5$ than 0.5 .

In summary, while the data is of high quality, the strong conclusions made in this paper are not substantiated by the data. The results showing an upward dispersion are new and there is likely comparisons to be made with the cuprates and pnictides like the authors have attempted. They have also done substantial work in understanding the field dependence which is commendable. However, due to the central interpretation of paramagnon scattering and the many unsubstantiated statements (list above) along with a poor comparison with previous works, I cannot recommend the paper for publication in its current form.

With our answers to the issues posed by the referee and the softened on paramagnon excitations we now make in the paper, we believe the paper is suitable for publication and hope that the referee will recommend it for publication.

Reviewer #3 (Remarks to the Author):

The authors have made several revisions from the previous version of the manuscript. The quality of the data remains high, and the addition of the new CeCoIn5 results has strengthened the experimental aspects of the work. Unfortunately my previous concerns about the interpretation are not really satisfied by the authors' response or by the revisions.

The authors continue to insist on two distinct scenarios for the resonance excitation: either excitons as found in weak coupling band + RPA models, or propagating spin-wave-like modes characteristic of a cooperative paramagnet --- black or white. The authors clearly show that their results are not consistent with a pure spin exciton as obtained from one particular weak coupling theory, and I accept this. However, the electronic system is very complex, with likely several

different f- and d-orbitals contributing to the magnetic response, and with the added complication of a quantum coherent superconducting state. With all this in the mix I feel it is over-simplistic to conclude that because there is an upwards curvature the spin resonance must be a paramagnon. There may well be some paramagnon character, but given the nature of the system, the susceptibility that is measured in the neutron spectra could derive from several different electronic contributions which cannot be separated in the experiment.

I would like to see the experimental results published because, as I said before, it is a very thorough study which makes a significant contribution to the field of unconventional superconductivity. I cannot recommend publication with the current strong conclusion unequivocally identifying the resonance as a paramagnon, but I think that if the authors were to considerably soften this conclusion while retaining the falsification of the spin excitation scenario then the paper would be suitable for publication.

We appreciate the positive comments from the referee, we now softened our claim and emphasize our results are inconsistent with the simple spin-exciton picture as the referee suggested.

In the following, we include our e-mail communication and raw data of Dr. Raymond for $E=0.4$ meV.

发件人:] [email correspondence redacted]

发送时间:

主题:

Replies to comments by the referee:

Reviewer #1 (Remarks to the Author):

“In my opinion, the authors have addressed all the criticism raised by all referees, and I recommend the paper for publication in its present form.”

We are grateful for the recommendation of the referee.

Reviewer #2 (Remarks to the Author):

“The paper is a resubmission based on my previous comments. The authors have softened some of their claims, but I still cannot recommend the paper for publication based on the points raised previously by myself and the other referees.

The authors still insist on heavily pushing the magnon picture with now the emphasis on their being the lack of any apparent downward dispersion. The authors don't have the resolution to make this claim and the results from Raymond et al are not conclusive either way. If there is no data, the authors cannot use this as a basis as stated in the abstract. I also find the new discussion regarding the field dependence highly speculative (although very interesting I have to admit).

In summary, I still cannot recommend the paper for publication. The paper is heavily biased and not substantiated by data.”

While we appreciate these criticisms from the referee, however, we disagree with them. First, we do not believe that we are “heavily pushing the magnon picture” as suggested by the referee. The emphasis of our paper is that the upward dispersion of the resonance is inconsistent with the spin-exciton picture, widely used to model the resonance mode in iron pnictides and cuprates. With this fact established, we then propose a possible magnon scenario which can explain our experimental observations. We DO NOT in any way suggest that the magnon scenario is conclusive or the only interpretation, but only offer it as a possible interpretation.

The referee's suggestion “If there is no data, the authors cannot use this as a basis as stated in the abstract.” is simply incorrect. Indeed (as stated in our previous resubmission), Dr. Raymond has agreed that in his raw data, there is no evidence for a downward dispersion. The results from two independent group show no signature of a downward dispersion, while an upward dispersion is clearly observed in our present work. We don't understand why the referee say there is no data. And the data unequivocally demonstrate the central point of our paper, the upward dispersing resonance cannot be interpreted as a spin-exciton. Finally, the referee states that “I also find the new discussion regarding the field dependence highly speculative”. First, we would like to point out that we did not perform any experiments in a magnetic field. Rather, we were asked by the referee to explain out result in the context of

the earlier experiments in a magnetic field. We have therefore performed additional theoretical studies, as discussed in our article, that have demonstrated the consistency between our results and results obtained in a magnetic field. As we have clearly stated, additional experiments need to be performed to confirm our proposed scenario. However, the referee seems to reject our paper because he/she does not like our explanation of an experiment performed by another group.

Reviewer #3 (Remarks to the Author):

“The authors have slightly softened their previous strong conclusion that the neutron spectrum supports a magnon interpretation. However, the manuscript still doesn't contain any discussion of the complexity of the magnetic response. In my view the revisions are still not adequate, and so regrettably I cannot recommend publication.”

We appreciate very much that the referee agrees that we have “softened” our claim of a magnon interpretation. We are sorry that the revised draft did not include additional discussion of the complexity of the magnetic response. In the revised draft, we have further softened our claim of the magnon interpretation. In addition, we now discuss explicitly the complexity of the magnetic response, and say that other interpretations of the data are possible. With these changes, we hope that the referee will now accept the paper for publication.